# The UK Chinese population with kidney failure: Clinical characteristics, management and access to kidney transplantation using 20 years of UK Renal Registry and NHS Blood and Transplant data

Katie Wong[1,2]*, Fergus J. Caskey[1,2‡], Anna Casula[3‡], Yoav Ben-Shlomo[1‡], Pippa Bailey[1,2]

1 Bristol Medical School: Population Health Sciences, University of Bristol, Bristol, United Kingdom,
2 Southmead Hospital, North Bristol NHS Trust, Bristol, United Kingdom, 3 UK Renal Registry, Bristol, United Kingdom

☉ These authors contributed equally to this work.
‡ FJC, AC and YBS also contributed equally to this work.
* Katie.Wong@nhs.net

**Data Availability Statement:** "No - some restrictions will apply" "The minimal underlying

## Abstract

### Background

Little is known about the clinical demographics of and access to transplantation for Chinese diaspora populations with kidney disease.

### Methods

The UK Renal Registry provided data on adults with ethnicity recorded as 'Chinese' or 'White' starting Kidney Replacement Therapy (KRT) 1/1/97-31/12/17. Baseline characteristics were compared between Chinese and White patients. Multivariable logistic regression models were used to investigate the relationships between Chinese ethnicity and i) being listed for deceased-donor transplantation at start of KRT, ii) being listed 2 years after start of KRT, iii) pre-emptive kidney transplantation, iv) kidney transplantation 3 years after start of KRT, and v) living-donor kidney transplantation (LDKT).

### Results

UK Chinese patients were younger at start of KRT (61.6 vs 65.6 years, p <0.001) and had more diabetic kidney disease (29% vs 20%, p<0.001) and glomerulonephritis (21% vs 13%, p<0.001) than White patients. We found evidence of interaction between ethnicity and sex. Compared to UK White men, UK Chinese men had lower odds of pre-emptive transplant (aOR 0.28, 95% CI [0.10–0.76]) and transplant within 3 years of KRT start (aOR 0.65, [95% CI 0.49–0.87], P = 0.004). UK White women and Chinese women had the same likelihood of pre-emptive transplant (aOR 0.78, 95% CI [0.38–1.61]), or transplant within 3 years of KRT

dataset cannot be included as it contains confidential, potentially identifying and sensitive patient information. The UK Kidney Association (UKKA) is the controller of the data and restricts sharing of individual level data if there is no direct relationship between the UKKA and the recipient. The data underlying these results are available at the UK Renal Registry (contact: ukrr-research@renalregistry.nhs.uk) for researchers who meet the criteria for access to confidential data."

**Funding:** This work was supported by the Elizabeth Blackwell Institute for Health Research, University of Bristol and the Wellcome Trust Institutional Strategic Support Fund, grant number 204813/Z/16/Z.

**Competing interests:** The authors have declared that no competing interests exist.

start (aOR 0.94, 95% CI [0.60–1.46]). Both UK Chinese men and women had markedly lower odds of LDKT compared to Whites aOR 0.34 [95% CI 0.21–0.53].

## Conclusions

UK Chinese are less likely to receive a LDKT. UK Chinese men have lower odds of accessing pre-emptive wait-listing and transplantation. Understanding whether these disparities reflect modifiable barriers will help ensure equitable access to transplantation.

## Introduction

The China Kidney Disease Network (CK-NET) and Hong Kong Renal Registry have reported the main cause of chronic kidney disease in their populations is diabetes [1, 2], but the causes of kidney disease in the Chinese diaspora have not been well described.

Data from the United Kingdom (UK) Renal Registry (UKRR) has shown that 0.5% of people with kidney failure in the UK are of Chinese ethnicity [3]. 393,141 (0.7%) of the UK population are UK Chinese, a number that increased 59% between 2001 and 2011 [4, 5].

Ethnic differences in the cause of kidney disease [6], dialysis rates [7, 8] and access to kidney transplantation [9] have been investigated across the UK White, South Asian and Black populations, however these have not been investigated in the UK Chinese population. These previous analyses have provided evidence of ethnic inequity in access to living-donor kidney transplantation (LDKT): both UKRR analyses [10] and the Access to Transplantation and Transplant Outcome Measures (ATTOM) study have reported that patients from ethnic minority groups are less likely to access LDKT [11], even after adjustment for socioeconomic status (SES). However, these studies have either excluded UK Chinese patients, or combined the UK Chinese with other minority ethnic groups for analysis due to small numbers. Combining ethnic minority populations for analysis reduces granularity of the ethnicity data and potentially may mask heterogeneity between different ethnic minority groups as regards (i) differential incidences of kidney disease and (ii) disparities in access to dialysis and transplant treatments between individual ethnic minority populations. The UK renal research strategy has called for research to better understand ethnic inequity and differential rates of kidney disease among ethnic minority groups [12]. To this end, we aimed to investigate the clinical characteristics and access to transplantation in the UK Chinese population with kidney disease.

## Materials and methods

Data on all incident adult (aged ≥18 years) patients starting kidney replacement therapy (KRT) was extracted for 20 years between 1/1/1997-31/12/2017 from centres submitting to the UKRR. Patients whose ethnicity was recorded as anything other than 'Chinese' or 'White', or with ethnicity data missing, were excluded. Patients who underwent kidney transplantation outside of the UK have been excluded. Data from centres that had poor completeness and quality of ethnicity data reporting were excluded. In keeping with UKRR reports, data from these centres were included in analysis from the year that they submitted ethnicity data for at least 70% of their patients. Patients were followed until death, censoring (recovery or lost to follow-up) or the end of the study period.

## Exposure, outcomes and confounders

Our main exposure was Chinese ethnicity (binary variable UK Chinese = 1 UK White = 0). We had five outcome measures in relation to access to transplantation. These were; i) being listed on the deceased-donor kidney transplant waiting list at start of KRT ii) being listed on the deceased-donor waiting list 2 years after start of KRT iii) pre-emptive kidney transplantation (from deceased or living donor), iv) kidney transplantation at 3 years after start of KRT, and v) living-donor kidney transplantation at any time. For analyses relating to access to transplantation and wait-listing on the UK National Kidney Allocation Scheme, patients ≥75 years at the age of start of KRT were excluded because of the high prevalence of comorbidity which decreases the likelihood of transplantation and the very small proportion of patients receiving a kidney transplant in the UK in this age group. Data on co-morbidity were incomplete, but patients who were recorded as having known malignancy at start of KRT were excluded from analyses investigating transplant access.

We obtained data on the following variables: age; sex; socioeconomic status (SES); primary renal diagnosis (PRD); time from first seeing a nephrologist to start of KRT. PRD was grouped using pre-2012 ERA-EDTA coding, as updated ERA-EDTA codes were unavailable for a proportion of patients (See S1 Table). The UKRR does not collect individual-level SES data. Instead, Index of Multiple Deprivation scores were used as an area level measure of SES. The Index of Multiple Deprivation (IMD) is a measure of relative deprivation for small areas (neighbourhoods). Each country in the UK (England, Wales, Scotland and Northern Ireland) has an IMD. The IMD ranks all small areas within a country from most deprived to least deprived (1 = most deprived) using different domains, which can include: Income Deprivation, Employment Deprivation, Education, Skills and Training Deprivation, Health Deprivation and Disability, Crime, Barriers to Housing and Services, Living Environment Deprivation. The relative ranking of a small area is it's IMD score (i.e. IMD score of 1 = most deprived) and these can be categorised into quintiles within each country (Quintile 1 = most deprived, Quintile 5 = least deprived). Each patient's country-specific (English [13], Welsh [14], Scottish [15], Northern Irish [16]) IMD quintile was derived using their postcode (zipcode).

## Statistical analyses

Means and standard deviations (SDs) were calculated for normally distributed continuous variables. Medians and interquartile ranges (IQRs) are presented for continuous variables whose distribution was not normal. Chi-square tests were used to compare categorical variables and Mann-Whitney U tests were used to compare medians. We identified missing data and investigated for any patterns of missingness. The proportion of missing data for categorical variables was compared for patients of UK Chinese and White ethnicities using Chi-Square tests, presented in S1 Table.

We performed and have presented complete case analyses. Specific variables (including Body Mass Index (BMI) and comorbidity data) had a high proportion of missing data (>50%) and so it was decided to exclude them from the analyses.

We used multivariable logistic regression models (Odds ratios (ORs), 95% Confidence Intervals (CIs) and p-values) to investigate the relationships between Chinese ethnicity (UK Chinese versus UK White) and i) being listed on the deceased-donor kidney transplant waiting list at start of KRT ii) being listed on the deceased-donor waiting list 2 years after start of KRT iii) pre-emptive kidney transplantation (from deceased or living donor), iv) kidney transplantation at 3 years after start of KRT, and v) living-donor kidney transplantation at any time. In addition for wait-listing at 2 years and transplant at 3 years we repeated the analysis using a competing-risks regression based on Fine and Gray's proportional subhazards model [17, 18] to account for the competing risk of death prior to either being wait-listed or transplantation.

All models were run unadjusted and then adjusted for the following covariates, specified *a priori*, age group (18–25 years, 26–35 years, 36–45 years, 46–55 years, 56–65 years, 66–74 years), PRD, sex and SES. PRD was managed as a categorical variable ('Diabetes', 'Glomerulonephritis' and 'Other'").

We tested for a priori interactions between Chinese ethnicity and the covariates age, sex and SES. When we found evidence of interaction we performed analyses stratified by the covariate. ORs with 95% CIs were calculated using robust standard errors to account for clustering by kidney centre. All statistical analyses were completed using STATA 15 software [19].

On behalf of the Renal Association, the UK Renal Registry (UKRR) collects patient data without consent with approval under section 251 of the NHS Act (2006), granted by the Health Research Authority's Confidentiality Advisory Group (ref 16/CAG/0064). Data are always pseudonymised prior to being analysed. The UKRR database has approval for research studies from the NHS North East Newcastle & North Tyneside 1 Research Ethics Committee (21/NE/0045). The UKRR is also permitted to share certain specified data items from its linked UKRR-NHSBT (NHS Blood and Transplant) dataset, including date of transplant and type of transplant received. The report was prepared with reference to the 'REporting of studies Conducted using Observational Routinely-collected health Data (RECORD)' Statement [20].

In the UK, all donation is overseen by NHS Blood and Transplant (NHSBT). For deceased donation, UK individuals can register as organ donors. In England, Wales, Scotland all adults are now considered to have 'deemed consent' or 'deemed authorisation' for organ donation: all adults are considered to have agreed to be an organ donor when they die unless they have a recorded decision not to donate, or are in an excluded group (those under the age of 18, people who lack the mental capacity to understand the new arrangements and take the necessary action, visitors to England, and those not living here voluntarily, people who have lived in England for less than 12 months before their death). In Northern Ireland unless individuals have registered as organ donors they are not considered to have agreed to donation. If an individual dies in circumstances in which organ donation is possible, an NHS specialist nurse will try to establish when a person is registered on the NHS Organ Donor Register. A patient's family/close friends/next of kin will then be asked to support a decision for organ donation to go ahead, and clinicians will not proceed if there are family objections.

All living donors have to be assessed to have capacity to donate, and to be donating free from coercion or payment by an Independent Assessor from the Human Tissue Authority. Payment for organ donation or facilitating organ donation is illegal. In March 2015, the UK signed the Council of Europe Convention against Trafficking in Human Organs www.declarationofistanbul.org/resources/recommended-reading/the-councilof-europe-convention-against-trafficking-in-human-organs.

All living kidney donors are consented according to local transplant centre protocols during their work-up process, and consent confirmed at time of donation.

The clinical and research activities reported are consistent with the Principles of the Declaration of Istanbul as outlined in the 'Declaration of Istanbul on Organ Trafficking and Transplant Tourism'.

## Results

### Clinical characteristics

The dataset comprised of 92,689 incident KRT patients. 0.5% (n = 492) were of Chinese ethnicity, 76% (n = 70,560) were White. The remaining 23.5% of patients were of Asian ethnicity (10%), Black (6%) Other (2%), or had ethnicity data missing (5%).

For BMI and Comorbidity missing data was >60% and so these variables were excluded from the analyses. Otherwise the proportion of missing data for covariates was small (≤5%) (S1 Table).

The characteristics of the UK Chinese KRT population compared to the UK White KRT population are shown in Table 1. Chinese patients were younger at start of KRT (61.6 vs 65.6 years, p <0.001) than White patients. There were marked differences in the causes of kidney failure: UK Chinese patients had more diabetic kidney disease (29% vs 20%, risk difference 0.09, 95% CI 0.05–0.13, p<0.001) and glomerulonephritis than White patients (21% vs 13%, risk difference 0.08, 95% CI 0.04–0.11, p<0.001).

Causes for glomerulonephritis in UK Chinese and White KRT patients are presented in Table 2.

## Management

There was moderate evidence that more Chinese patients started KRT on peritoneal dialysis that White patients (26% vs 23%, p = 0.02). There was no evidence of difference between the

**Table 1. Characteristics of UK Chinese and UK White KRT populations.**

| | Chinese n = 492 | White n = 70,560 | p value |
|---|---|---|---|
| Median age at start of KRT, years (IQR) | 61.6 (47.7–72.1) | 65.6 (51.8–75.2) | <0.001 ** |
| Sex, n (%) | | | 0.31 * |
| Male | 399 (61) | 44,442 (63) | |
| Female | 193 (39) | 26,118 (37) | |
| Missing | 0 (0) | 0 (0) | |
| IMD quintile, n (%) | | | 0.61 * |
| 1 Most deprived | 112 (23) | 17, 585 (25) | |
| 2 | 119 (24) | 15, 381 (22) | |
| 3 | 97 (20) | 13,810 (20) | |
| 4 | 90 (18) | 12,418 (18) | |
| 5 Least deprived | 73 (15) | 11,083 (16) | |
| Missing | 1 (0) | 283 (0) | |
| Cause of renal failure, n (%) | | | <0.001* |
| Diabetes | 141 (29) | 14,105 (20) | |
| Glomerulonephritis | 103 (21) | 9,213 (13) | |
| Hypertension | 33 (7) | 4,207 (6) | |
| Polycystic kidney disease | 27 (5) | 5,434 (8) | |
| Pyelonephritis | 15 (3) | 5,494 (8) | |
| Renovascular disease | 10 (2) | 5,122 (7) | |
| Uncertain | 101 (21) | 13,055 (19) | |
| Other | 43 (9) | 11,691(17) | |
| Missing | 19 (4) | 2,239 (3) | |
| Treatment at start KRT, n (%) | | | 0.02* |
| Haemodialysis | 346 (70) | 50,100 (71) | |
| Peritoneal dialysis | 128 (26) | 15,964 (23) | |
| Transplant | 18 (4) | 4,496 (6) | |
| Missing | 0 (0) | 0 (0) | |
| Median time from first seen by nephrologist to start of KRT, days (IQR) | 784 (154, 1985) | 826 (164, 2063) | 0.60**a |

*Chi[2] test;

**Mann-Whitney U test

KRT = Kidney Replacement Therapy

a Completed case analysis.

**Table 2. Causes of glomerulonephritis in the UK Chinese and White populations.**

|  | Chinese n (%) | White n (%) | P value[*] |
|---|---|---|---|
| **Glomerulonephritis- no histology** | 8 (8) | 1266 (14) | 0.005 |
| **Focal Segmental Glomerulosclerosis with nephrotic syndrome in children** | 0 (0) | 101 (1) |  |
| **Focal Segmental Glomerulosclerosis with nephrotic syndrome in adults** | 11 (11) | 1037 (11) |  |
| **IgA disease** | 55 (53) | 3096 (34) |  |
| **Dense deposit disease** | 0 (0) | 85 (1) |  |
| **Membranous nephropathy** | 9 (9) | 918 (10) |  |
| **Membranoproliferative glomerulonephritis** | 7 (7) | 504 (5) |  |
| **Crescentic glomerulonephritis** | 1 (1) | 504 (5) |  |
| **Glomerulonephritis with biopsy in adults** | 12 (12) | 1706 (19) |  |
| **Total** | 103 | 9213 |  |

[*]Fisher's exact test.

UK Chinese KRT population and the White KRT population in median days from first seen by nephrologist to start of KRT KRT (784 vs 826, p = 0.60).

## Access to transplantation

The findings of the multivariable logistic regression analyses are presented in Table 3.

Even after adjustment for potential confounders, UK Chinese patients had lower odds of being waitlisted at the start of KRT (adjusted OR (aOR) 0.74, [95% CI 0.56–0.98], P = 0.03) but were as likely to be waitlisted at 2 years (aOR 1.26, [95% CI 1.00–1.60], P = 0.06) compared to White patients. The results from the competing risk model were almost identical (subdistribution hazard ratio (SHR) 1.19 [1.00–1.41] P = 0.05). UK Chinese individuals were also less likely to receive a pre-emptive kidney transplant (aOR 0.49, [95% CI 0.30–0.81], P = 0.005) and less likely to be transplanted within 3 years of starting KRT (aOR 0.65, [95% CI 0.49–0.87], P = 0.004) compared to White patients. Chinese patients were 66% less likely to receive a LDKT than White patients (aOR 0.34, 95% CI [0.21–0.53], P<0.001). Adjusted competing risk analyses and cause-specific proportional hazard ratios are presented in S2 Table and were very similar to the results from the logistic models.

There was evidence of statistical interaction between Chinese ethnicity and sex for the outcome variables of access to pre-emptive transplant (likelihood ratio test = 0.05), and access to transplant at 3 years (likelihood ratio test p = 0.04): Chinese men were less likely than White

**Table 3. Results of multivariable logistic regression analyses investigating Chinese ethnicity and access to waitlisting and transplantation.**

|  | Unadjusted analysis | Adjusted analysis[*] |
|---|---|---|
|  | Chinese vs White | Chinese vs White |
|  | OR, (95% CI), | OR, (95% CI), |
|  | P-value | P-value |
| Waitlisted at KRT start | 0.78 (0.59–1.02), P = 0.07 | 0.74 (0.56–0.98), P = 0.03 |
| Waitlisted at 2 years | 1.29 (1.06–1.58), p = 0.01 | 1.26 (1.0–1.6), p = 0.06 |
| Pre-emptive transplant | 0.52 (0.32–0.84), P = 0.007 | 0.49 (0.30–0.81), P = 0.005 |
| Transplanted within 3 years of KRT start | 0.74 (0.58–0.93), P = 0.01 | 0.65 (0.49–0.87), P = 0.004 |
| Living donor kidney transplant | 0.34 (0.22–0.52), P<0.001 | 0.34 (0.21–0.53), P<0.001 |

[*]Adjusted for age, sex, primary renal disease, and socioeconomic status.

men to have a pre-emptive transplant (aOR 0.28, 95% CI [0.10–0.76]), or be transplanted within 3 years of KRT start (aOR 0.50, 95%CI [0.34–0.73]). However Chinese women appeared to be as likely as White women to have a pre-emptive transplant (aOR 0.78, 95% CI [0.38–1.61]), or to be transplanted within 3 years (aOR 0.94, 95% CI [0.60–1.46]).

## Discussion

To our knowledge, this study is the first to describe renal disease and access to transplantation in the UK Chinese population, and one of the first to describe renal disease in the Chinese diaspora. We found a greater burden of diabetes in the UK Chinese KRT population compared to the White KRT population: this may reflect a high prevalence of diabetes in the UK Chinese, or a greater pre-disposition to diabetic nephropathy in Chinese individuals with diabetes as compared to Whites and requires further investigation. The prevalence of diabetes in the UK Chinese was reported as 3.8% in men and 3.3% in women [21] in 2004, but may have increased subsequently, as has been reported in China [22, 23] and Hong Kong [24], where prevalence of diabetes has risen and is reported as 11% and 10% respectively. Specific variations in RAGE [25], ELMO1 [26] and TGF-β1 [27] genes, and their gene-environment interactions, have been found to be associated with diabetic nephropathy in the Han Chinese population, but have not been investigated in the Chinese diaspora, where environmental exposures may be different. The Chinese Kidney Disease Network (CK-NET) has reported a 15.1% prevalence of glomerulonephritis in their CKD population, 10.6% of which were diagnosed with IgA nephropathy, a number which had fallen from 19% in 2010 [1], which is lower than the 54% prevalence of IgA nephropathy in patients with glomerulonephritis in this UK study. The reasons behind this require further investigation.

The slightly increased rates of PD in UK Chinese patients may be associated with relatively lower average BMIs in the Chinese population [28]. The quantity of missing BMI data in the UKRR dataset prevented investigation of this association in this study. Hong Kong have had a successful "PD first" policy for more than 30 years, with 72.9% of their dialysis population being treated with PD [29]. It may be that familiarity with this mode of kidney replacement therapy has led to increased uptake of PD in the UK Chinese KRT population. Additionally, studies from Hong Kong have suggested that high transporter status is less common [30] and lower dialysis volumes [31] are required in their cohort. It would be beneficial to elucidate whether this is also the case in the UK Chinese KRT population, and if so, peritoneal dialysis encouraged.

Our findings suggest that UK Chinese men are less likely to receive a pre-emptive kidney transplant, or to receive a transplant 3 years after start of KRT than UK White men; UK Chinese women are no less likely to receive a pre-emptive transplant, or to receive a transplant 3 years after start of KRT than UK White women. However, both UK Chinese men and women are less likely to receive a LDKT.

Whether these disparities reflect modifiable barriers at health system, clinician or patient level requires further investigation. We found no evidence that the UK Chinese population (both men and women) with kidney failure are presenting late to nephrology services and thus precluding timely transplant work-up, with no significant difference in time from first seen by nephrologist to start of KRT between UK Chinese and White patients. Our finding of no difference in the subdistribution hazard of wait-listing at 2 years, after accounting for the competing risk of death before wait-listing, for UK Chinese patients (both men and women) compared to White patients suggests that fitness for transplantation is not a significant barrier; however the UK Chinese may require more time to be assessed as suitable. Multicentre cross-sectional studies have demonstrated that estimated glomerular filtration rate (eGFR)

calculations [32, 33], in particular the Modification of Diet in Renal Disease study (MDRD) equation can overestimate GFR when compared to cystatin C based GFR measurements in a Chinese cohort. However another Chinese cohort study found that eGFR calculations under-estimated GFR compared to gold standard iohexol measurements, especially in diabetic patients [34]. Therefore, inaccuracy of standard eGFR calculations, validated in White and Black patients, may also be hindering accurate assessment of kidney function in the UK Chinese.

Sex disparity in access to kidney transplantation (lower in women than men on KRT) has previously been observed in the UK and worldwide [35–37], due in part to pregnancy induced HLA sensitisation. It has been suggested that ethnic minority women are particularly affected due to being more likely to be multiparous [38, 39]. A questionnaire based study examining sex disparity in an African-American haemodialysis cohort further found that women were less likely to have discussions around and be evaluated for a kidney transplant than men [40]. However, our findings suggest that UK Chinese women are no less likely to be pre-emptively listed or receive a kidney transplant at 3 years after starting KRT than White women; whether UK Chinese women are less sensitized, have a cultural advantage (e.g. increased care-giver support) or differences in attitude towards kidney transplantation requires further investigation. A population based retrospective cohort study found lower mortality from breast and cervical cancer in Ontarian Canadian Chinese immigrant women compared to the general population [41], even when removing the "healthy immigrant" effect. A potential reason suggested was that Chinese women were more likely to pursue aggressive anti-neoplastic treatments due to cultural differences in treatment philosophy. Our findings highlight that barriers to kidney transplantation may be individual to certain ethnic minority groups; research combining ethnic minority populations together for analysis may fail to appreciate this heterogeneity.

The reasons why the UK Chinese KRT population, both men and women, are less likely than White patients to receive a LDKT are not well understood. A qualitative study examining the views of Chinese immigrants to Canada [42] towards both deceased and living organ donation reported the importance of "filial piety" in this cohort- the concept of respect for parents and elders. Whilst this study described a potential conflict between the desire to return the body whole in death to parents and ancestors as an act of respect and living or deceased organ donation, it also suggested a willingness to consider living donation amongst interviewees, especially from parent to child. A survey in a Malaysian Chinese cohort similarly reported concerns around the sanctity of the body in death [43]. However, this has not been investigated in other Chinese diaspora, and previous studies have shown that attitudes towards organ donation can change depending on host country [44].

## Strengths and limitations

This study utilised national UK registry and NHSBT data to investigate a previously unstudied group: the UK Chinese population with renal failure. The UKRR is one of few renal registries internationally to collect individual level ethnicity data [45], which enabled us to describe the clinical demographics and access to transplantation in this more specific ethnic group, which may not have been possible elsewhere. However, this study has a number of limitations: i) The data regarding ethnicity were physician entered, and may not match how the ethnic group with which the patient identifies ii) Data on patient BMI and co-morbidity were incomplete and prevented the use of these variables in our analysis. In particular we were unable to ascertain the prevalence of co-morbidities which may have precluded transplantation in the study cohort. (iii) The number of UK Chinese people in the study was relatively small.

The reasons for the observed disparity in wait-listing and access to deceased donor transplantation between UK Chinese men and women requires further investigation. Future research should use qualitative methods with the UK Chinese population to elucidate whether there are cultural or information barriers to kidney donation within this population, or attitudes which may favour kidney transplantation, in particular focused on understanding whether there are differences between UK Chinese men and women.

Whether the differential rates of kidney disease seen in the UK Chinese in our study is also the case in other Chinese diaspora patient groups also requires further study.

In summary, our study has found evidence that the UK Chinese KRT population differs from the White KRT population. Although the number of UK Chinese with kidney failure is small, we have found evidence that these individuals, in particular UK Chinese men, are disadvantaged in access to wait-listing and transplantation. Early identification and support of this population may therefore be beneficial.

## Supporting information

**S1 Table. Missing data analysis.**
(DOCX)

**S2 Table. Competing risk analyses.**
(DOCX)

**S3 Table. Results of multivariable logistic regression analyses investigating Chinese ethnicity and access to wait-listing and transplantation, stratified by sex.**
(DOCX)

**S1 File.**
(PDF)

## Author Contributions

**Conceptualization:** Katie Wong, Fergus J. Caskey, Pippa Bailey.

**Data curation:** Katie Wong, Anna Casula.

**Formal analysis:** Katie Wong, Fergus J. Caskey, Anna Casula, Yoav Ben-Shlomo, Pippa Bailey.

**Funding acquisition:** Katie Wong.

**Investigation:** Katie Wong, Pippa Bailey.

**Methodology:** Katie Wong, Fergus J. Caskey, Anna Casula, Pippa Bailey.

**Supervision:** Fergus J. Caskey, Yoav Ben-Shlomo, Pippa Bailey.

**Writing – original draft:** Katie Wong, Pippa Bailey.

**Writing – review & editing:** Katie Wong, Fergus J. Caskey, Yoav Ben-Shlomo, Pippa Bailey.

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
