## [Decision Letter · Decision Letter 0]

12 Jul 2021

PONE-D-21-05646

The UK Chinese population with kidney failure: Clinical characteristics, management and access to kidney transplantation using 20 years of UK Renal Registry and NHS Blood and Transplant Data

PLOS ONE

Dear Dr. Wong,

Thank you for submitting your manuscript to PLOS ONE. After careful consideration, we feel that it has merit but does not fully meet PLOS ONE’s publication criteria as it currently stands. Therefore, we invite you to submit a revised version of the manuscript that addresses the points raised during the review process.

The Editor reviewed the manuscript and recommends several changes and corrections. Please read specific comments. 

Reviewer # 1:

The paper by Wong et al goes over the differences in causes of renal failure, access to transplantation and other characteristics in UK White and UK Chinese patient populations. The paper appears well written and the study was methodologically sound. It is surprising that statistically significant differences were found when the Chinese cohort was only about 500 patients in size and adjustment by regression confounders was applied. There is certainly no reason to question the presented results, though I speculate that applying more conservative testing with verification of logistic regression’s assumptions (which almost everyone seems to skip nowadays) would lead to problems related to sample size. Good points were made regarding higher rates of diabetes and possible need for a special method for GFR calculation in Chinese patients

Reviewer # 2:

Please address the following points:

The analysis is a comparison of Chinese and White populations listed in the UK registry system. Please provide details of exclusions (percentages) for Chinese and White populations.

Please explain more precisely the category as “being listed on the deceased donor waiting list for 2 years” and “kidney transplantation at 3 years after start of KRT”. Does the first category mean to be listed at least 2 years and more? Does the kidney transplantation category mean kidney transplantation within 3 years? Please clarify these terms. Also, pre-emptive transplantation is from deceased or live donor?

What mean “data on co-morbidity were incomplete”.

Please explain “The UKRR does not collect individual-level SES data. Index of Multiple Deprivation (IMD) scores were used as an area level measure of SES” as this statement is not clear. The authors collect and use SES data but then say that such data are not collected. Please clarify and explain better SES data collection.

Please explain Index of Multiple Deprivation (IMD) scores as an average reader is not familiar with this system. It is also not clear what is most deprived vs. least deprived and differences among coutries. Please explain.  How the authors calculated IMD for missing data? The statement is that IMD for England was used. Please explain.

The statistical analysis is not clear for missing data “We identified missing data and investigated for any patterns of missingness.”

The statement “Intervals (CIs) and p-values) to investigate the relationships between Chinese ethnicity (UK Chinese versus UK White) and the five outcomes listed above” is not clear. Where is listed above?

Please confirm the statement “The UK Renal Registry has been granted a section 251 exemption by the Health Research Authority, allowing the sharing and processing of identifiable patient information from kidney units without individual patient consent.”

The Results section provide that “. 0.5% (n=501) were of Chinese ethnicity, 76% (n=70,575) were White”. What is the remaining percentage? Please clarify.

Please explain how in Table 3, the analysis was adjusted for SES for cases that have missing data? The Materials and Methods state that there are many missing data.  

Please provide more attention for the results showing differences between males and females. What is the reason for the difference in Chinese males to be so significantly different than Chinese females? Please confirm or refute the assumption that all differences revealed in this manuscript are refer exclusively to the Chinese males. Please try to explain this unique observation by possible different variables.

The Discussion section has conclusions about the entire Chinese population (males and females) whereas the results showed to be concentrated in Chinese males. Please clarify and make additional analyses to make sure what conclusions refer to the entire Chinese population (males and females with the same rates in reported observations) vs. conclusions exclusively found in Chinese male population. Please make very clear what observations are in which Chinese groups.

The limitation section should be more specific as well as the conclusion section needs to reflect the documented observations regarding gender disparities.

The authors need to confirm the entire analysis in Chinese male vs. Chinese female population. The differences at different levels of analyses need to be clearly indicated. The Results and Discussion sections need to be adjusted.

We look forward to receiving your revised manuscript.

Kind regards,

Stanislaw Stepkowski

Academic Editor

PLOS ONE

Additional Editor Comments (if provided):

The Editor reviewed the manuscript and recommends several changes and corrections. Please read specific comments.

Reviewer # 1:

The paper by Wong et al goes over the differences in causes of renal failure, access to transplantation and other characteristics in UK White and UK Chinese patient populations. The paper appears well written and the study was methodologically sound. It is surprising that statistically significant differences were found when the Chinese cohort was only about 500 patients in size and adjustment by regression confounders was applied. There is certainly no reason to question the presented results, though I speculate that applying more conservative testing with verification of logistic regression’s assumptions (which almost everyone seems to skip nowadays) would lead to problems related to sample size. Good points were made regarding higher rates of diabetes and possible need for a special method for GFR calculation in Chinese patients

Reviewer # 2:

Please address the following points:

The analysis is a comparison of Chinese and White populations listed in the UK registry system. Please provide details of exclusions (percentages) for Chinese and White populations.

Please explain more precisely the category as “being listed on the deceased donor waiting list for 2 years” and “kidney transplantation at 3 years after start of KRT”. Does the first category mean to be listed at least 2 years and more? Does the kidney transplantation category mean kidney transplantation within 3 years? Please clarify these terms. Also, pre-emptive transplantation is from deceased or live donor?

What mean “data on co-morbidity were incomplete”.

Please explain “The UKRR does not collect individual-level SES data. Index of Multiple Deprivation (IMD) scores were used as an area level measure of SES” as this statement is not clear. The authors collect and use SES data but then say that such data are not collected. Please clarify and explain better SES data collection.

Please explain Index of Multiple Deprivation (IMD) scores as an average reader is not familiar with this system. It is also not clear what is most deprived vs. least deprived and differences among coutries. Please explain. How the authors calculated IMD for missing data? The statement is that IMD for England was used. Please explain.

The statistical analysis is not clear for missing data “We identified missing data and investigated for any patterns of missingness.”

The statement “Intervals (CIs) and p-values) to investigate the relationships between Chinese ethnicity (UK Chinese versus UK White) and the five outcomes listed above” is not clear. Where is listed above?

Please confirm the statement “The UK Renal Registry has been granted a section 251 exemption by the Health Research Authority, allowing the sharing and processing of identifiable patient information from kidney units without individual patient consent.”

The Results section provide that “. 0.5% (n=501) were of Chinese ethnicity, 76% (n=70,575) were White”. What is the remaining percentage? Please clarify.

Please explain how in Table 3, the analysis was adjusted for SES for cases that have missing data? The Materials and Methods state that there are many missing data.

Please provide more attention for the results showing differences between males and females. What is the reason for the difference in Chinese males to be so significantly different than Chinese females? Please confirm or refute the assumption that all differences revealed in this manuscript are refer exclusively to the Chinese males. Please try to explain this unique observation by possible different variables.

The Discussion section has conclusions about the entire Chinese population (males and females) whereas the results showed to be concentrated in Chinese males. Please clarify and make additional analyses to make sure what conclusions refer to the entire Chinese population (males and females with the same rates in reported observations) vs. conclusions exclusively found in Chinese male population. Please make very clear what observations are in which Chinese groups.

The limitation section should be more specific as well as the conclusion section needs to reflect the documented observations regarding gender disparities.

The authors need to confirm the entire analysis in Chinese male vs. Chinese female population. The differences at different levels of analyses need to be clearly indicated. The Results and Discussion sections need to be adjusted

Journal Requirements:

2. Thank you for including your ethics statement:  "The UK Renal Registry has been granted a section 251 exemption by the Health Research Authority, allowing the sharing and processing of identifiable patient information from kidney units without individual patient consent".   

a. Please amend your current ethics statement to include the full name of the ethics committee/institutional review board(s) that  reviewed and approved the UK Renal Registry research programme. Please include relevant approval numbers. 

In addition, please provide additional information about the patient records/samples used in your retrospective study, including whether all data were fully anonymized before you accessed them, and also state whether the ethics committee that approved the research programme waived the need for written informed consent.

3. We note that your study involved tissue/organ transplantation. Please provide the following information regarding tissue/organ donors for transplantation cases analyzed in your study.

a. Please provide the source(s) of the transplanted tissue/organs used in the study, including the institution name and a non-identifying description of the donor(s).

b. Please state in your response letter and ethics statement whether the transplant cases for this study involved any vulnerable populations; for example, tissue/organs from prisoners, subjects with reduced mental capacity due to illness or age, or minors.

- If a vulnerable population was used, please describe the population, justify the decision to use tissue/organ donations from this group, and clearly describe what measures were taken in the informed consent procedure to assure protection of the vulnerable group and avoid coercion. 

- If a vulnerable population was not used, please state in your ethics statement, “None of the transplant donors was from a vulnerable population and all donors or next of kin provided written informed consent that was freely given.”

c. In the Methods, please provide detailed information about the procedure by which informed consent was obtained from organ/tissue donors or their next of kin. In addition, please provide a blank example of the form used to obtain consent from donors, and an English translation if the original is in a different language.

d. Please indicate whether the donors were previously registered as organ donors. If tissues/organs were obtained from deceased donors or cadavers, please provide details as to the donors’ cause(s) of death.

e. Please provide the participant recruitment dates and the period during which transplant procedures were done (as month and year).

f. Please discuss whether medical costs were covered or other cash payments were provided to the family of the donor. If so, please specify the value of this support (in local currency and equivalent to U.S. dollars).

This research is supported by the University of Bristol Elizabeth Blackwell Institute Clinical Primer award, which receives a contribution from the Wellcome Institute Strategic Support Fund. PB, FJC, AC, Y-BS have nothing to disclose. 

This research is supported by the University of Bristol Elizabeth Blackwell Institute Clinical Primer award, which receives a contribution from the Wellcome Institute Strategic Support Fund. PB, FJC, AC, Y-BS have nothing to disclose. 

6. Please amend either the abstract on the online submission form (via Edit Submission) or the abstract in the manuscript so that they are identical.

7. We note you have included a table to which you do not refer in the text of your manuscript. Please ensure that you refer to Table 2 in your text; if accepted, production will need this reference to link the reader to the Table.

9.  We noticed you have some minor occurrence of overlapping text with the following previous publication(s), which needs to be addressed:

- https://www.journaltocs.ac.uk/index.php?action=tocs&journalISSN=1460-2385

-https://www.era-edta.org/VirtualCongress2020/Accepted_Abstracts_ERAEDTA_2020.pdf

-https://academic.oup.com/ndt/article/35/Supplement_3/gfaa142.P0772/5853027

In your revision ensure you cite all your sources (including your own works), and quote or rephrase any duplicated text outside the methods section. Further consideration is dependent on these concerns being addressed.

Reviewers' comments:

Reviewer's Responses to Questions

**Comments to the Author**

1. Is the manuscript technically sound, and do the data support the conclusions?

Reviewer #1: Yes

Reviewer #2: Yes

2. Has the statistical analysis been performed appropriately and rigorously? 

Reviewer #1: Yes

Reviewer #2: Yes

3. Have the authors made all data underlying the findings in their manuscript fully available?

Reviewer #1: Yes

Reviewer #2: Yes

4. Is the manuscript presented in an intelligible fashion and written in standard English?

Reviewer #1: Yes

Reviewer #2: Yes

5. Review Comments to the Author

Reviewer #1: The paper by Wong et al goes over the differences in causes of renal failure, access to transplantation and other characteristics in UK White and UK Chinese patient populations. The paper appears well written and the study was methodologically sound. It is surprising that statistically significant differences were found when the Chinese cohort was only about 500 patients in size and adjustment by regression confounders was applied. There is certainly no reason to question the presented results, though I speculate that applying more conservative testing with verification of logistic regression’s assumptions (which almost everyone seems to skip nowadays) would lead to problems related to sample size. Good points were made regarding higher rates of diabetes and possible need for a special method for GFR calculation in Chinese patients.

Reviewer #2: Please address the following points:

The analysis is a comparison of Chinese and White populations listed in the UK registry system. Please provide details of exclusions (percentages) for Chinese and White populations.

Please explain more precisely the category as “being listed on the deceased donor waiting list for 2 years” and “kidney transplantation at 3 years after start of KRT”. Does the first category mean to be listed at least 2 years and more? Does the kidney transplantation category mean kidney transplantation within 3 years? Please clarify these terms. Also, pre-emptive transplantation is from deceased or live donor?

What mean “data on co-morbidity were incomplete”.

Please explain “The UKRR does not collect individual-level SES data. Index of Multiple Deprivation (IMD) scores were used as an area level measure of SES” as this statement is not clear. The authors collect and use SES data but then say that such data are not collected. Please clarify and explain better SES data collection.

Please explain Index of Multiple Deprivation (IMD) scores as an average reader is not familiar with this system. It is also not clear what is most deprived vs. least deprived and differences among coutries. Please explain. How the authors calculated IMD for missing data? The statement is that IMD for England was used. Please explain.

The statistical analysis is not clear for missing data “We identified missing data and investigated for any patterns of missingness.”

The statement “Intervals (CIs) and p-values) to investigate the relationships between Chinese ethnicity (UK Chinese versus UK White) and the five outcomes listed above” is not clear. Where is listed above?

Please confirm the statement “The UK Renal Registry has been granted a section 251 exemption by the Health Research Authority, allowing the sharing and processing of identifiable patient information from kidney units without individual patient consent.”

The Results section provide that “. 0.5% (n=501) were of Chinese ethnicity, 76% (n=70,575) were White”. What is the remaining percentage? Please clarify.

Please explain how in Table 3, the analysis was adjusted for SES for cases that have missing data? The Materials and Methods state that there are many missing data.

Please provide more attention for the results showing differences between males and females. What is the reason for the difference in Chinese males to be so significantly different than Chinese females? Please confirm or refute the assumption that all differences revealed in this manuscript are refer exclusively to the Chinese males. Please try to explain this unique observation by possible different variables.

The Discussion section has conclusions about the entire Chinese population (males and females) whereas the results showed to be concentrated in Chinese males. Please clarify and make additional analyses to make sure what conclusions refer to the entire Chinese population (males and females with the same rates in reported observations) vs. conclusions exclusively found in Chinese male population. Please make very clear what observations are in which Chinese groups.

The limitation section should be more specific as well as the conclusion section needs to reflect the documented observations regarding gender disparities.

The authors need to confirm the entire analysis in Chinese male vs. Chinese female population. The differences at different levels of analyses need to be clearly indicated. The Results and Discussion sections need to be adjusted.

6. PLOS authors have the option to publish the peer review history of their article (what does this mean?). If published, this will include your full peer review and any attached files.

Reviewer #1: **Yes: **Dulat Bekbolsynov

Reviewer #2: No

---

## [Author Response · Author response to Decision Letter 0]

25 Nov 2021

Please also find the below response in the uploaded file "Response to Reviewers" 

Dear Editors, 

Many thanks for your review of our manuscript “The UK Chinese population with kidney failure: Clinical characteristics, management and access to kidney transplantation using 20 years of UK Renal Registry and NHS Blood and Transplant Data”, and the opportunity to respond to reviewer’s comments and improve our manuscript. 

Please find our responses to points raised below, with reviewer comments in bold: 

Reviewer 1: 

The paper by Wong et al goes over the differences in causes of renal failure, access to transplantation and other characteristics in UK White and UK Chinese patient populations. The paper appears well written and the study was methodologically sound. It is surprising that statistically significant differences were found when the Chinese cohort was only about 500 patients in size and adjustment by regression confounders was applied. There is certainly no reason to question the presented results, though I speculate that applying more conservative testing with verification of logistic regression’s assumptions (which almost everyone seems to skip nowadays) would lead to problems related to sample size. Good points were made regarding higher rates of diabetes and possible need for a special method for GFR calculation in Chinese patients 

We thank the reviewer for their time reviewing this manuscript, and for their positive comments regarding the discussion. 

Reviewer 2: 

Thank you for taking the time to review our paper, and for your detailed response. Please find answers to individual points below. 

1) The analysis is a comparison of Chinese and White populations listed in the UK registry system. Please provide details of exclusions (percentages) for Chinese and White populations. 

All patients whose ethnicity was recorded by the UK Renal Registry as ‘Chinese’ or ‘White’ were included. 

2) Please explain more precisely the category as “being listed on the deceased donor waiting list for 2 years” and “kidney transplantation at 3 years after start of KRT”. Does the first category mean to be listed at least 2 years and more? Does the kidney transplantation category mean kidney transplantation within 3 years? Please clarify these terms. Also, pre-emptive transplantation is from deceased or live donor? 

“Being listed on the deceased-donor kidney transplant waiting list 2 years after start of KRT” 

This outcome category includes all individuals with a date of being listed on the deceased-donor kidney transplant waiting list within 2 years, or ≤730 days, of their date of start of Kidney Replacement Therapy (KRT). 

“Kidney transplantation at 3 years after start of KRT” 

This outcome category includes all individuals with a date of kidney transplantation within 3 years, or ≤1095 days, of their date of start of KRT. 

“Pre-emptive transplantation” 

Many thanks for bringing this to our attention. This outcome category includes all individuals who received a kidney transplant prior to starting another form of KRT, and includes individuals receiving deceased and living donor kidney transplant. Additional information has been added in the Exposures, outcomes and confounders section of the Methods to clarify: “iii) pre-emptive kidney transplantation (from deceased or living donor)” page 5, line 16. 

3) What mean “data on co-morbidity were incomplete”. 

The UK Renal Registry collects data on an individual’s co-morbidities, including ischaemic heart disease and malignancy. However, these data were incomplete: as stated in the results section under “Clinical Characteristics”, missing data was >60%. 

4) Please explain “The UKRR does not collect individual-level SES data. Index of Multiple Deprivation (IMD) scores were used as an area level measure of SES” as this statement is not clear. The authors collect and use SES data but then say that such data are not collected. Please clarify and explain better SES data collection. 

Please explain Index of Multiple Deprivation (IMD) scores as an average reader is not familiar with this system. It is also not clear what is most deprived vs. least deprived and differences among coutries. Please explain.  How the authors calculated IMD for missing data? The statement is that IMD for England was used. Please explain. 

Many thanks for highlighting the need for further clarity in this section. It has now been amended in the manuscript to: 

“The UKRR does not collect individual-level SES data. Instead, Index of Multiple Deprivation scores were used as an area level measure of SES. The Index of Multiple Deprivation (IMD) is a measure of relative deprivation for small areas (neighbourhoods). Each country in the UK (England, Wales, Scotland and Northern Ireland) has an IMD. The IMD ranks all small areas within a country from most deprived to least deprived (1=most deprived) using different domains, which can include: Income Deprivation, Employment Deprivation, Education, Skills and Training Deprivation, Health Deprivation and Disability, Crime, Barriers to Housing and Services, Living Environment Deprivation. The relative ranking of a small area is it’s IMD score (i.e. IMD score of 1= most deprived) and these can be categorised into quintiles within each country.(Quintile 1= most deprived, Quintile 5= least deprived). 

Each patient’s country-specific (English(13), Welsh(14), Scottish(15), Northern Irish(16)) IMD quintile was derived using their postcode (zipcode).” Page 6, lines 4-16. 

Data were analysed assuming that any patient with IMD score data but missing country of residence data, lived in England. However, on review, no patient in the dataset had an IMD score without also having country of residence data. The sentence “For patients missing data on country of residence, the patient’s IMD quintile was calculated using the IMD quintile parameters for England, as the majority of patients were from this nation” has therefore been deleted for clarity. 

5) The statistical analysis is not clear for missing data “We identified missing data and investigated for any patterns of missingness.” 

Thank you for bringing this to our attention. 

This has now been amended to “We identified missing data and investigated for any patterns of missingness. The proportion of missing data for categorical variables was compared for patients of UK Chinese and White ethnicities using Chi-Square tests, presented in S1 Tables of Supplementary Materials.” (Page 6, lines 22-25) 

6) The statement “Intervals (CIs) and p-values) to investigate the relationships between Chinese ethnicity (UK Chinese versus UK White) and the five outcomes listed above” is not clear. Where is listed above? 

The five outcomes investigated are listed in the section Exposure, outcomes and confounders: “We had five outcome measures in relation to access to transplantation. These were; i) being listed on the deceased-donor kidney transplant waiting list at start of KRT ii) being listed on the deceased-donor waiting list 2 years after start of KRT iii) pre-emptive kidney transplantation (from deceased or living donor), iv) kidney transplantation at 3 years after start of KRT, and v) living-donor kidney transplantation at any time”. 

For clarity, these five outcomes have now been listed again in the Statistical Analyses section (page 7, lines 6-10). 

7) Please confirm the statement “The UK Renal Registry has been granted a section 251 exemption by the Health Research Authority, allowing the sharing and processing of identifiable patient information from kidney units without individual patient consent.” 

This statement has been clarified following request from the journal editors. It now reads: “On behalf of the Renal Association, the UK Renal Registry (UKRR) collects patient data without consent with approval under section 251 of the NHS Act (2006), granted by the Health Research Authority’s Confidentiality Advisory Group (ref 16/CAG/0064). Data are always pseudonymised prior to being analysed. The UKRR database has approval for research studies from the NHS North East Newcastle & North Tyneside 1 Research Ethics Committee (21/NE/0045)”. (Page 7 Lines 23-34 and Page 8 Lines 1-6) 

8) The Results section provide that “. 0.5% (n=501) were of Chinese ethnicity, 76% (n=70,575) were White”. What is the remaining percentage? Please clarify. 

The results section has now been amended to: “0.5% (n=501) were of Chinese ethnicity, 76% (n=70,575) were White. The remaining 23.5% of patients were of Asian ethnicity (10%), Black (6%) Other (2%), or had ethnicity data missing (5%) (Page 10, Lines 4-5). 

Please explain how in Table 3, the analysis was adjusted for SES for cases that have missing data? The Materials and Methods state that there are many missing data.   

The Materials and Methods does not refer to many missing data for this particular variable. The number of patients missing SES/IMD data, as presented in Table 1, for UK Chinese (n=1) and UK White (n=283) patients was small (<1% for both groups). For patients with IMD missing data, completed case analysis was undertaken. 

9) Please provide more attention for the results showing differences between males and females. What is the reason for the difference in Chinese males to be so significantly different than Chinese females? Please confirm or refute the assumption that all differences revealed in this manuscript are refer exclusively to the Chinese males. Please try to explain this unique observation by possible different variables. 

The Discussion section has conclusions about the entire Chinese population (males and females) whereas the results showed to be concentrated in Chinese males. Please clarify and make additional analyses to make sure what conclusions refer to the entire Chinese population (males and females with the same rates in reported observations) vs. conclusions exclusively found in Chinese male population. Please make very clear what observations are in which Chinese groups. 

The limitation section should be more specific as well as the conclusion section needs to reflect the documented observations regarding gender disparities.The authors need to confirm the entire analysis in Chinese male vs. Chinese female population. The differences at different levels of analyses need to be clearly indicated. The Results and Discussion sections need to be adjusted. 

Thank you for highlighting this interesting finding. 

Statistical interactions between Chinese ethnicity and Sex were found for the outcome variables of (i) Pre-emptive transplant, and (ii) Transplantation within 3 years of KRT start, such that only UK Chinese men were less likely to receive a pre-emptive transplant, or a transplant within 3 years of KRT start compared to UK White men. 

No evidence of a statistical interaction between Chinese ethnicity and Sex was found for the associations with outcome variables Waitlisting at KRT start, Waitlisting at 2 years, or Living donor kidney transplantation. 

An additional table presenting results of multivariable logistic regression analyses investigating Chinese ethnicity and access to wait-listing and transplantation, stratified by Sex, has been added in S3 Tables in Supplementary Materials for clarity. Existing S2 Tables in supplementary materials presents competing risks models investigating the association between access to wait-listing and transplantation at 3 years, competing risk of death before wait-listing or transplantation, stratified by Sex. 

These findings have also been made more explicit in the Discussion section which now reads: 

“Our findings suggest that UK Chinese men are less likely to receive a pre-emptive kidney transplant, or to receive a transplant 3 years after start of KRT than UK White men; UK Chinese women are no less likely to receive a pre-emptive transplant, or to receive a transplant 3 years after start of KRT than UK White women. However, both UK Chinese men and women are less likely to receive a LDKT. (Page 15, Lines 18-22)” Where results discussed include both UK Chinese men and women, the Discussion has been amended to include “(both men and women)” (Page 16, lines 1 and 5, Page 17 line 9). 

The reasons for this disparity between UK Chinese men and women is currently unclear and requires further investigation. Alongside this UK renal registry analysis, I am also undertaking qualitative interviews examining the views of the UK Chinese towards kidney transplantation from both deceased and living donor which aims to gain a deeper understanding of these results, for instance to understand whether UK Chinese men encounter additional cultural, social or medical barriers to pre-emptive transplantation or transplantation within 3 years of starting KRT. 

I have amended the Strengths and Limitations to reflect this: 

“The reasons for the observed disparity in wait-listing and access to deceased donor transplantation between UK Chinese men and women requires further investigation. Future research should use qualitative methods with the UK Chinese population to elucidate whether there are cultural or information barriers to kidney donation within this population, or attitudes which may favour kidney transplantation, in particular focused on understanding whether there are differences between UK Chinese men and women. (Page 18, lines 8-15)”. 

Journal Requirements: 

Many thanks for taking the time to review this paper. 

1) Please ensure that your manuscript meets PLOS ONE's style requirement 

The manuscript has now been amended to meet PLOS ONE’s style requirements. 

2) Thank you for including your ethics statement:  "The UK Renal Registry has been granted a section 251 exemption by the Health Research Authority, allowing the sharing and processing of identifiable patient information from kidney units without individual patient consent".    

a. Please amend your current ethics statement to include the full name of the ethics committee/institutional review board(s) that reviewed and approved the UK Renal Registry research programme. Please include relevant approval numbers.  

In addition, please provide additional information about the patient records/samples used in your retrospective study, including whether all data were fully anonymized before you accessed them, and also state whether the ethics committee that approved the research programme waived the need for written informed consent. 

Ethics statement has been amended in both the manuscript and online submission to: 

“On behalf of the Renal Association, the UK Renal Registry (UKRR) collects patient data without consent with approval under section 251 of the NHS Act (2006), granted by the Health Research Authority’s Confidentiality Advisory Group (ref 16/CAG/0064). Data are always pseudonymised prior to being analysed. The UKRR database has approval for research studies from the NHS North East Newcastle & North Tyneside 1 Research Ethics Committee (21/NE/0045)” (Page 7, lines 23-24 and Page 8, lines 1-6). 

3. We note that your study involved tissue/organ transplantation. Please provide the following information regarding tissue/organ donors for transplantation cases analyzed in your study. 

a. Please provide the source(s) of the transplanted tissue/organs used in the study, including the institution name and a non-identifying description of the donor(s). 

b. Please state in your response letter and ethics statement whether the transplant cases for this study involved any vulnerable populations; for example, tissue/organs from prisoners, subjects with reduced mental capacity due to illness or age, or minors. 

c. In the Methods, please provide detailed information about the procedure by which informed consent was obtained from organ/tissue donors or their next of kin. In addition, please provide a blank example of the form used to obtain consent from donors, and an English translation if the original is in a different language. 

d. Please indicate whether the donors were previously registered as organ donors. If tissues/organs were obtained from deceased donors or cadavers, please provide details as to the donors’ cause(s) of death. 

e. Please provide the participant recruitment dates and the period during which transplant procedures were done (as month and year). 

f. Please discuss whether medical costs were covered or other cash payments were provided to the family of the donor. If so, please specify the value of this support (in local currency and equivalent to U.S. dollars). 

a. Please provide the source(s) of the transplanted tissue/organs used in the study, including the institution name and a non-identifying description of the donor(s). 

This study was undertaken using retrospective data from the UK Renal Registry (UKRR) and NHS Blood and Transplant. Data from the UKRR is provided for research fully pseudonymised as per its section 251 approval granted by the Health Research Authority’s Confidentiality Advisory Group.  It is therefore not possible to identify the donors for individual patients in this cohort. 

However, the great majority of kidney transplants reported in this registry study were undertaken in the UK. Of 501 UK Chinese patients who received a kidney transplant between 1/1/97-31/12/17, only 9 patients (5.2% of total) received kidney transplants abroad: 7 patients within 3 years of KRT start, 2 patients after 3 years of KRT start. Of 70,575 UK White patients who received a kidney transplant in this time period, only 15 patients (0.1% of total) received kidney transplants abroad: 12 patients within 3 years of KRT start, 3 patients after 3 years of KRT start. 

In the UK, all donation is overseen by NHS Blood and Transplant (NHSBT). All living donors have to be assessed to have capacity to donate, and to be donating free from coercion or payment by an Independent Assessor from the Human Tissue Authority. Payment for organ donation or facilitating organ donation is illegal. In March 2015, the UK signed the Council of Europe Convention against Trafficking in Human Organs www.declarationofistanbul.org/resources/recommended-reading/the-councilof-europe-convention-against-trafficking-in-human-organs. Therefore all transplant activity in the UK as reported in this paper adheres to the Declaration of Istanbul. 

The manuscript has now been amended such that the small number of patients transplanted outside of the UK have been excluded in analyses as the UKRR holds no information as to the providence of the organs transplanted. 

This has been updated in the methods section to read: “Patients who underwent kidney transplantation outside of the UK have been excluded from analyses” (Page 5, Line 5) 

b. Please state in your response letter and ethics statement whether the transplant cases for this study involved any vulnerable populations; for example, tissue/organs from prisoners, subjects with reduced mental capacity due to illness or age, or minors. 

The ethics statement has now been amended to read: 

“In the UK, all donation is overseen by NHS Blood and Transplant (NHSBT). All living donors have to be assessed to have capacity to donate, and to be donating free from coercion or payment by an Independent Assessor from the Human Tissue Authority. Payment for organ donation or facilitating organ donation is illegal. In March 2015, the UK signed the Council of Europe Convention against Trafficking in Human Organs www.declarationofistanbul.org/resources/recommended-reading/the-councilof-europe-convention-against-trafficking-in-human-organs. Therefore all transplant activity in the UK as reported in this paper adheres to the Declaration of Istanbul. All living donors in England, Wales and Northern Ireland must be more that 18 years old at time of donation, in Scotland the legal lower age limit is 16 years old.” 

c. In the Methods, please provide detailed information about the procedure by which informed consent was obtained from organ/tissue donors or their next of kin. In addition, please provide a blank example of the form used to obtain consent from donors, and an English translation if the original is in a different language. 

This study was undertaken using retrospective UKRR and NHSBT data. The UKRR collates data received from UK renal centres. We as the authors and the UK Renal Registry were therefore not directly involved with identifying or obtaining informed consent from donors or recipients. 

However in the UK, consent for donation of organs in overseen by NHSBT. The following statement has been added to the Methods section: 

“In the UK, all donation is overseen by NHS Blood and Transplant (NHSBT). For deceased donation, UK individuals can register as organ donors. In England, Wales, Scotland all adults are now considered to have ‘deemed consent’ or ‘deemed authorisation’ for organ donation: all adults are considered to have agreed to be an organ donor when they die unless they have a recorded decision not to donate, or are in an excluded group (those under the age of 18, people who lack the mental capacity to understand the new arrangements and take the necessary action, visitors to England, and those not living here voluntarily, people who have lived in England for less than 12 months before their death). In Northern Ireland unless individuals have registered as organ donors they are not considered to have agreed to donation. If an individual dies in circumstances in which organ donation is possible, an NHS specialist nurse will try to establish when a person is registered on the NHS Organ Donor Register. A patient’s family/close friends/next of kin will then be asked to support a decision for organ donation to go ahead, and clinicians will not proceed if there are family objections. 

All living donors have to be assessed to have capacity to donate, and to be donating free from coercion or payment by an Independent Assessor from the Human Tissue Authority. Payment for organ donation or facilitating organ donation is illegal. In March 2015, the UK signed the Council of Europe Convention against Trafficking in Human Organs www.declarationofistanbul.org/resources/recommended-reading/the-councilof-europe-convention-against-trafficking-in-human-organs. 

All living kidney donors are consented according to local transplant centre protocols during their work-up process, and consent confirmed at time of donation”. (Page 8, Lines 9-25 and Page 9, Lines 1-8) 

A blank example of a consent form used has been uploaded. 

d. Please indicate whether the donors were previously registered as organ donors. If tissues/organs were obtained from deceased donors or cadavers, please provide details as to the donors’ cause(s) of death. 

In the UK, organs can be donated after death even if an individual has not been registered as an organ donor. In practice, this is only undertaken with consent from a potential donor’s family and next of kin. 

Donor cause of death was not relevant to the analyses undertaken in this study. 

Individuals receiving organs from donors outside of the UK, where protocols for donor consent cannot been verified, have now been removed (please see response 3a). 

e. Please provide the participant recruitment dates and the period during which transplant procedures were done (as month and year). 

Patients included in this study were transplanted between 1/1/97-31/12/17. 

f. Please discuss whether medical costs were covered or other cash payments were provided to the family of the donor. If so, please specify the value of this support (in local currency and equivalent to U.S. dollars). 

These data are not collected by the UKRR. 

The NHS in the UK is a publicly funded healthcare system that is free to users at the point of access. Kidney donation incurs no direct medical costs to the donor. However, NHS England commissioning policy for reimbursement of living donors aims to ensure that the financial impact on living donors is cost neutral: it is founded on the premise that there should be no financial incentive or disincentive in becoming a living donor. It specifies that travel expenses will be reimbursed by the NHS for all living donors (from UK or overseas), with additional loss of earnings, child care, employment costs and accommodation reimbursement considered on an individual basis (https://www.england.nhs.uk/wp-content/uploads/2018/08/Policy-Reimbursement-of-Expenses-for-Living-Donors-Updated-August-2018.pdf). 

4. We note that the grant information you provided in the ‘Funding Information’ and ‘Financial Disclosure’ sections do not match.  

5. Please include your amended funding statements within your cover letter; we will change the online submission form on your behalf. 

The amended funding statement is included in a cover letter and in this response to reviewers, as below: 

“This work was supported by the Elizabeth Blackwell Institute for Health Research, University of Bristol and the Wellcome Trust Institutional Strategic Support Fund, grant number 204813/Z/16/Z.” 

6. Please amend either the abstract on the online submission form (via Edit Submission) or the abstract in the manuscript so that they are identical. 

This has now been amended. 

7. We note you have included a table to which you do not refer in the text of your manuscript. Please ensure that you refer to Table 2 in your text; if accepted, production will need this reference to link the reader to the Table. 

Thank you for highlighting this, Table 2 has now been referenced in the main text (Page 10, Lines 15-16). 

8. Please include captions for your Supporting Information files at the end of your manuscript, and update any in-text citations to match accordingly. 

Captions have now been included for Supporting Information files. 

9.  We noticed you have some minor occurrence of overlapping text with the following previous publication(s), which needs to be addressed: 

1) https://www.journaltocs.ac.uk/index.php?action=tocs&journalISSN=1460-2385

2) https://www.era-edta.org/VirtualCongress2020/Accepted_Abstracts_ERAEDTA_2020.pdf

3) https://academic.oup.com/ndt/article/35/Supplement_3/gfaa142.P0772/5853027

In your revision ensure you cite all your sources (including your own works), and quote or rephrase any duplicated text outside the methods section. Further consideration is dependent on these concerns being addressed. 

The first (1) publication referred to is a website (journaltocs.ac.uk) which updates with new journal articles regularly, with this link being for articles from Nephrology Dialysis and Transplantation. I do not recognise any of these papers and the site updates articles regularly- would it be possible get some further clarification as to which paper the overlap refers to? 

The last 2 links, (2) and (3) are two links to the abstract for an earlier draft of this work, which was submitted to the ERA-EDTA conference and accepted for poster presentation. Poster abstracts were published in an NDT supplement. It may have been that (1) refers to the same abstract publications prior to the website updating. Please let me know if this abstract needs referencing in the main manuscript.

---

## [Editor Report · Decision Letter 1]

9 Feb 2022

The UK Chinese population with kidney failure: Clinical characteristics, management and access to kidney transplantation using 20 years of UK Renal Registry and NHS Blood and Transplant Data

PONE-D-21-05646R1

Dear Dr. Wong,

We’re pleased to inform you that your manuscript has been judged scientifically suitable for publication and will be formally accepted for publication once it meets all outstanding technical requirements.

Kind regards,

Stanislaw Stepkowski

Academic Editor

PLOS ONE
---

## [Editor Report · Acceptance letter]

15 Feb 2022

PONE-D-21-05646R1 

The UK Chinese population with kidney failure: Clinical characteristics, management and access to kidney transplantation using 20 years of UK Renal Registry and NHS Blood and Transplant Data 

Dear Dr. Wong:

I'm pleased to inform you that your manuscript has been deemed suitable for publication in PLOS ONE. Congratulations! Your manuscript is now with our production department. 

Kind regards, 

on behalf of

Dr. Stanislaw Stepkowski 

Academic Editor

PLOS ONE